# Unsupervised Scale-consistent Depth and Ego-motion Learning from Monocular Video

**Jia-Wang Bian**[1,2]**, Zhichao Li**[3]**, Naiyan Wang**[3]**, Huangying Zhan**[1,2]

**Chunhua Shen**[1,2]**, Ming-Ming Cheng**[4]**, Ian Reid**[1,2]

[1]University of Adelaide, Australia
[2]Australian Centre for Robotic Vision, Australia
[3]TuSimple, China
[4]Nankai University, China

## Abstract

Recent work has shown that CNN-based depth and ego-motion estimators can be learned using unlabelled monocular videos. However, the performance is limited by unidentified moving objects that violate the underlying static scene assumption in geometric image reconstruction. More significantly, due to lack of proper constraints, networks output scale-inconsistent results over different samples, *i.e.*, the ego-motion network cannot provide full camera trajectories over a long video sequence because of the per-frame scale ambiguity. This paper tackles these challenges by proposing a geometry consistency loss for scale-consistent predictions and an induced self-discovered mask for handling moving objects and occlusions. Since we do not leverage multi-task learning like recent works, our framework is much simpler and more efficient. Comprehensive evaluation results demonstrate that our depth estimator achieves the state-of-the-art performance on the KITTI dataset. Moreover, we show that our ego-motion network is able to predict a globally scale-consistent camera trajectory for long video sequences, and the resulting visual odometry accuracy is competitive with the recent model that is trained using stereo videos. To the best of our knowledge, this is the first work to show that deep networks trained using unlabelled monocular videos can predict globally scale-consistent camera trajectories over a long video sequence.

## 1   Introduction

Depth and ego-motion estimation is crucial for various applications in robotics and computer vision. Traditional methods are usually hand-crafted stage-wise systems, which rely on correspondence search [1, 2] and multi-view geometry [3, 4] for estimation. Recently, deep learning based methods [5, 6] show that the depth can be inferred from a single image by using Convolutional Neural Network (CNN). Especially, unsupervised methods [7–11] show that CNN-based depth and ego-motion networks can be solely trained on monocular video sequences without using ground-truth depth or stereo image pairs (pose supervision). The principle is that one can warp the image in one frame to another frame using the predicted depth and ego-motion, and then employ the image reconstruction loss as the supervision signal [7] to train the network. However, the performance limitation arises due to the moving objects that violate the underlying static scene assumption in geometric image reconstruction. More significantly, due to lack of proper constraints the network predicts scale-

inconsistent results over different samples, *i.e.*, the ego-motion network cannot provide a full camera trajectory over a long video sequence because of the per-frame scale ambiguity[1].

To the best of our knowledge, no previous work (unsupervised learning from monocular videos) addresses the scale-inconsistency issue mentioned above. To this end, we propose a geometry consistency loss for tackling the challenge. Specifically, for any two consecutive frames sampled from a video, we convert the predicted depth map in one frame to 3D space, then project it to the other frame using the estimated ego-motion, and finally minimize the inconsistency of the projected and the estimated depth maps. This explicitly enforces the depth network to predict geometry-consistent (of course scale-consistent) results over consecutive frames. With iterative sampling and training from videos, depth predictions on each consecutive image pair would be scale-consistent, and the frame-to-frame consistency can eventually propagate to the entire video sequence. As the scale of ego-motions is tightly linked to the scale of depths, the proposed ego-motion network can predict scale-consistent relative camera poses over consecutive snippets. We show that just simply accumulating pose predictions can result in globally scale-consistent camera trajectories over a long video sequence (Fig. 3).

Regarding the challenge of moving objects, recent work addresses it by introducing an additional optical flow [9–11, 13] or semantic segmentation network [14]. Although this improves performance significantly, it also brings about huge computational cost during training. Here we show that we could automatically discover a mask from the proposed geometry consistency term for solving the problem without introducing new networks. Specifically, we can easily locate pixels that belong to dynamic objects/occluded regions or difficult regions (*e.g.*, textureless regions) using the proposed term. By assigning lower weights to those pixels, we can avoid their impact to the fragile image reconstruction loss (see Fig. 2 for mask visualization). Compared with these recent approaches [9–11] that leverage multi-task learning, the proposed method is much simpler and more efficient.

We conduct detailed ablation studies that clearly demonstrate the efficacy of the proposed approach. Furthermore, comprehensive evaluation results on the KITTI [15] dataset show that our depth network outperforms state-of-the-art models that are trained in more complicated multi-task learning frameworks [9–11, 16]. Meanwhile, our ego-motion network is able to predict scale-consistent camera trajectories over long video sequences, and the accuracy of trajectory is competitive with the state-of-the-art model that is trained using stereo videos [17].

To summarize, our main contributions are three-fold:

- We propose a geometry consistency constraint to enforce the scale-consistency of depth and ego-motion networks, leading to a globally scale-consistent ego-motion estimator.
- We propose a self-discovered mask for dynamic scenes and occlusions by the aforementioned geometry consistency constraint. Compared with other approaches, our proposed approach does not require additional optical flow or semantic segmentation networks, which makes the learning framework simpler and more efficient.
- The proposed depth estimator achieves state-of-the-art performance on the KITTI dataset, and the proposed ego-motion predictor shows competitive visual odometry results compared with the state-of-the-art model that is trained using stereo videos.

## 2 Related work

Traditional methods rely on the disparity between multiple views of a scene to recover the 3D scene geometry, where at least two images are required [3]. With the rapid development of deep learning, Eigen et al. [5] show that the depth can be predicted from a single image using Convolution Neural Network (CNN). Specifically, they design a coarse-to-fine network to predict the single-view depth and use the ground truth depths acquired by range sensors as the supervision signal to train the network. However, although these supervised methods [5, 6, 18–21] show high-quality flow and depth estimation results, it is expensive to acquire ground truth in real-world scenes.

Without requiring the ground truth depth, Garg et al. [22] show that a single-view depth network can be trained using stereo image pairs. Instead of using depth supervision, they leverage the established epipolar geometry [3]. The color inconsistency between a left image and a synthesized left image warped from the right image is used as the supervision signal. Following this idea, Godard et al. [23] propose to constrain the left-right consistency for regularization, and Zhan et al. [17] extend the method to stereo videos. However, though stereo pairs based methods do not require the ground truth depth, accurately rectifying stereo cameras is also non-trivial in real-world scenarios.

To that end, Zhou et al. [7] propose a fully unsupervised framework, in which the depth network can be learned solely from monocular videos. The principle is that they introduce an additional ego-motion network to predict the relative camera pose between consecutive frames. With the estimated depth and relative pose, image reconstruction as in [22] is applied and the photometric loss is used as the supervision signal. However, the performance is limited due to dynamic objects that violate the underlying static scene assumption in geometric image reconstruction. More importantly, Zhou et al. [7]'s method suffers from the per-frame scale ambiguity, in that a single and consistent scaling of the camera translations is missing and only direction is known. As a result, the ego-motion network cannot predict a full camera trajectory over a long video sequence.

For handling moving objects, recent work [9, 10] proposes to introduce an additional optical flow network. Even more recently [11] introduces an extra motion segmentation network. Although they show significant performance improvement, there is a huge additional computational cost added into the basic framework, yet they still suffer from the scale-inconsistency issue. Besides, Liu et al. [24] use depth projection loss for supervision density, similar to the proposed consistency loss, but their method relies on the pre-computed 3D reconstruction for supervision.

To the best of our knowledge, this paper is the first one to show that the ego-motion network trained in monocular videos can predict a globally scale-consistent camera trajectory over a long video sequence. This shows significant potentials to leverage deep learning methods in Visual SLAM [12] for robotics and autonomous driving.

## 3 Unsupervised Learning of Scale-consistent Depth and Ego-motion

### 3.1 Method Overview

Our goal is to train depth and ego-motion networks using monocular videos, and constrain them to predict scale-consistent results. Given two consecutive frames $(I_a, I_b)$ sampled from an unlabeled video, we first estimate their depth maps $(D_a, D_b)$ using the depth network, and then predict the relative 6D camera pose $P_{ab}$ between them using the pose network.

With the predicted depth and relative camera pose, we can synthesize the reference image $I'_a$ by interpolating the source image $I_b$ [25, 7]. Then, the network can be supervised by the photometric loss between the real image $I_a$ and the synthesized one $I'_a$. However, due to dynamic scenes that violate the geometric assumption in image reconstruction, the performance of this basic framework is limited. To this end, we propose a geometry consistency loss $L_{GC}$ for scale-consistency and a self-discovered mask $M$ for handling the moving objects and occlusions. Fig. 1 shows an illustration of the proposed loss and mask.

Our overall objective function can be formulated as follows:

$$L = \alpha L_p^M + \beta L_s + \gamma L_{GC}, \tag{1}$$

where $L_p^M$ stands for the weighted photometric loss ($L_p$) by the proposed mask $M$, and $L_s$ stands for the smoothness loss. We train the network in both forward and backward directions to maximize the data usage, and for simplicity we only derive the loss for the forward direction.

In the following sections, we first introduce the widely used photometric loss and smoothness loss in Sec. 3.2, and then describe the proposed geometric consistency loss in Sec. 3.3 and the self-discovered mask in Sec. 3.4.

### 3.2 Photometric loss and smoothness loss

**Photometric loss.** Leveraging the brightness constancy and spatial smoothness priors used in classical dense correspondence algorithms [26], previous works [7, 9–11] have used the photometric

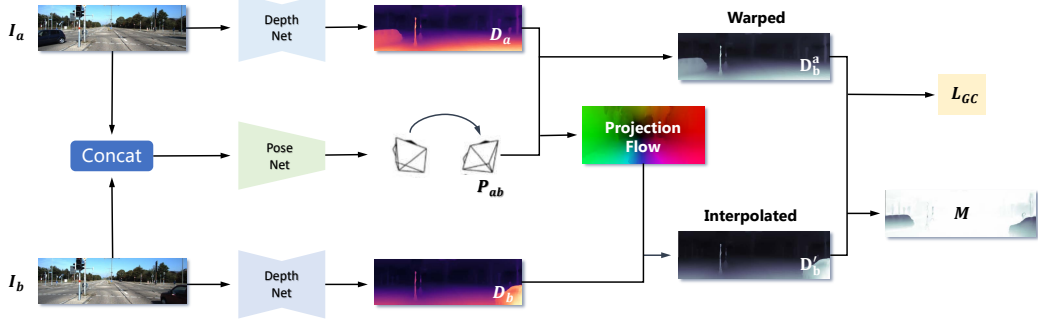

Figure 1: Illustration of the proposed geometry consistency loss and self-discover mask. Given two consecutive frames $(I_a, I_b)$, we first estimate their depth maps $(D_a, D_b)$ and relative pose $(P_{ab})$ using the network, then we get the warped $(D_b^a)$ by converting $D_a$ to 3D space and projecting to the image plane of $I_b$ using $P_{ab}$, and finally we use the inconsistency between $D_b^a$ and the $D_b'$ interpolated from $D_b$ as the geometric consistency loss $L_{GC}$ (Eqn. 6) to supervise the network training. Here, we interpolate $D_b$ because the projection flow does not lie on the pixel grid of $I_b$. Besides, we discover a mask $M$ (Eqn. 7) from the inconsistency map for handling dynamic scenes and ill-estimated regions (Fig. 2). For clarity, the photometric loss and smoothness loss are not shown in this figure.

error between the warped frame and the reference frame as an unsupervised loss function for training the network.

With the predicted depth map $D_a$ and the relative camera pose $P_{ab}$, we synthesize $I_a'$ by warping $I_b$, where differentiable bilinear interpolation [25] is used as in [7]. With the synthesized $I_a'$ and the reference image $I_a$, we formulate the objective function as

$$L_p = \frac{1}{|V|} \sum_{p \in V} \|I_a(p) - I_a'(p)\|_1, \tag{2}$$

where $V$ stands for valid points that are successfully projected from $I_a$ to the image plane of $I_b$, and $|V|$ defines the number of points in $V$. We choose $L_1$ loss due to its robustness to outliers. However, it is still not invariant to illumination changes in real-world scenarios. Here we add an additional image dissimilarity loss SSIM [27] for better handling complex illumination changes, since it normalizes the pixel illumination. We modify the photometric loss term Eqn. 2 as:

$$L_p = \frac{1}{|V|} \sum_{p \in V} (\lambda_i \|I_a(p) - I_a'(p)\|_1 + \lambda_s \frac{1 - \text{SSIM}_{aa'}(p)}{2}), \tag{3}$$

where $\text{SSIM}_{aa'}$ stands for the element-wise similarity between $I_a$ and $I_a'$ by the SSIM function [27]. Following [23, 9, 11], we use $\lambda_i = 0.15$ and $\lambda_s = 0.85$ in our framework.

**Smoothness loss.** As the photometric loss is not informative in low-texture nor homogeneous region of the scene, existing work incorporates a smoothness prior to regularize the estimated depth map. We adopt the edge-aware smoothness loss used in [11], which is formulated as:

$$L_s = \sum_p (e^{-\nabla I_a(p)} \cdot \nabla D_a(p))^2, \tag{4}$$

where $\nabla$ is the first derivative along spatial directions. It ensures that smoothness is guided by the edge of images.

### 3.3 Geometry consistency loss

As mentioned before, we enforce the geometry consistency on the predicted results. Specifically, we require that $D_a$ and $D_b$ (related by $P_{ab}$) conform the same 3D scene structure, and minimize their differences. The optimization not only encourages the geometry consistency between samples in a batch but also transfers the consistency to the entire sequence. e.g., depths of $I_1$ agree with depths of $I_2$ in a batch; depths of $I_2$ agree with depths of $I_3$ in another training batch. Eventually, depths

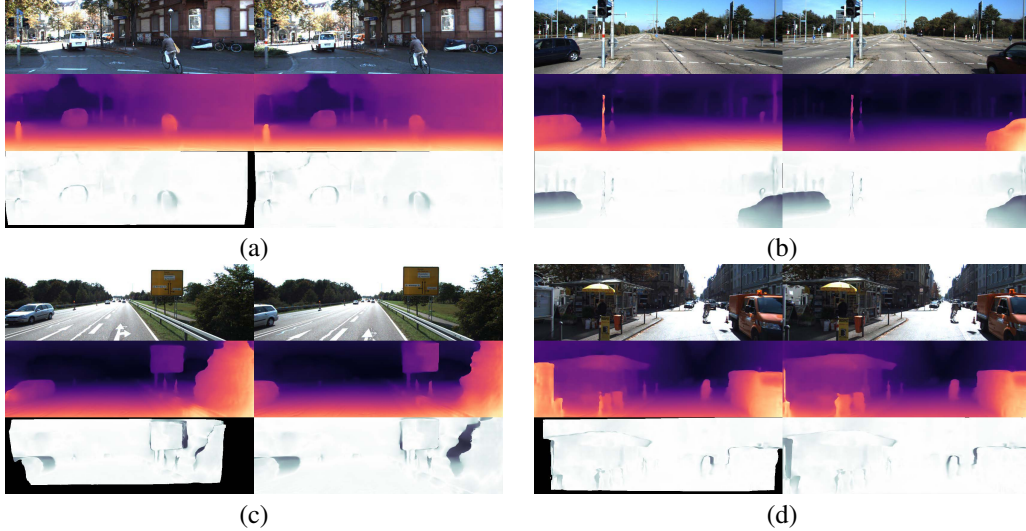

Figure 2: Visual results. Top to bottom: sample image, estimated depth, self-discovered mask. The proposed mask can effectively identify occlusions and moving objects.

of $I_i$ of a sequence should all agree with each other. As the pose network is naturally coupled with the depth network during training, our method yields scale-consistent predictions over the entire sequence.

With this constraint, we compute the depth inconsistency map $D_{\text{diff}}$. For each $p \in V$, it is defined as:

$$D_{\text{diff}}(p) = \frac{|D_b^a(p) - D_b'(p)|}{D_b^a(p) + D_b'(p)} \tag{5}$$

where $D_b^a$ is the computed depth map of $I_b$ by warping $D_a$ using $P_{ab}$, and $D_b'$ is the interpolated depth map from the estimated depth map $D_b$ (Note that we cannot directly use $D_b$ because the warping flow does not lie on the pixel grid ). Here we normalize their difference by their sum. This is more intuitive than the absolute distance as it treats points at different absolute depths equally in optimization. Besides, the function is symmetric and the outputs are naturally ranging from 0 to 1, which contributes to numerical stability in training.

With the inconsistency map, we simply define the proposed geometry consistency loss as:

$$L_{GC} = \frac{1}{|V|} \sum_{p \in V} D_{\text{diff}}(p), \tag{6}$$

which minimizes the geometric distance of predicted depths between each consecutive pair and enforces their scale-consistency. With training, the consistency can propagate to the entire video sequence. Due to the tight link between ego-motion and depth predictions, the ego-motion network can eventually predict globally scale-consistent trajectories (Fig. 3).

### 3.4 Self-discovered mask

To handle moving objects and occlusions that may impair the network training, recent work propose to introduce an additional optical flow [9–11] or semantic segmentation network [14]. This is effective, however it also introduces extra computational cost and training burden. Here, we show that these regions can be effectively located by the proposed inconsistency map $D_{\text{diff}}$ in Eqn. 5.

There are several scenarios that result in inconsistent scene structure observed from different views, including (1) dynamic objects, (2) occlusions, and (3) inaccurate predictions for difficult regions. Without separating them explicitly, we observe each of these will result in $D_{\text{diff}}$ increasing from its ideal value of zero.

Based on this simple observation, we propose a weight mask $M$ as $D_{\text{diff}}$ is in $[0, 1]$:

$$M = 1 - D_{\text{diff}}, \tag{7}$$

which assigns low/high weights for inconsistent/consistent pixels. It can be used to re-weight the photometric loss. Specifically, we modify the photometric loss in Eqn. 3 as

$$L_p^M = \frac{1}{|V|} \sum_{p \in V} (M(p) \cdot L_p(p)). \qquad (8)$$

By using the mask, we mitigate the adverse impact from moving objects and occlusions. Further, the gradients computed on inaccurately predicted regions carry less weight during back-propagation. Fig. 2 shows visual results for the proposed mas, which coincides with our anticipation stated above.

## 4 Experiment

### 4.1 Implementation details

**Network architecture.**  For the depth network, we experiment with DispNet [7] and DispRes-Net [11], which takes a single RGB image as input and outputs a depth map. For the ego-motion network, PoseNet without the mask prediction branch [7] is used. The network estimates a 6D relative camera pose from a concatenated RGB image pair. Instead of computing the loss on multiple-scale outputs of the depth network (4 scales in [7] or 6 scales in [11]), we empirically find that using single-scale supervision (*i.e.*, only compute the loss on the finest output) is better (Tab. 4). Our single-scale supervision not only improves the performance but also contributes a more concise training pipeline. We hypothesize the reason of this phenomenon is that the photometric loss is not accurate in low-resolution images, where the pixel color is over-smoothed.

**Single-view depth estimation.**  The proposed learning framework is implemented using PyTorch Library [28]. For depth network, we train and test models on KITTI raw dataset [15] using Eigen [5]'s split that is the same with related works [10, 9, 11, 7]. Following [7], we use a snippet of three sequential video frames as a training sample, where we set the second image as reference frame to compute loss with other two images and then inverse their roles to compute loss again for maximizing the data usage. The data is also augmented with random scaling, cropping and horizontal flips during training, and we experiment with two input resolutions ($416 \times 128$ and $832 \times 256$). We use ADAM [29] optimizer, and set the batch size to 4 and the learning rate to $10^{-4}$. During training, we adopt $\alpha = 1.0$, $\beta = 0.1$, and $\gamma = 0.5$ in Eqn. 1. We train the network in 200 epochs with 1000 randomly sampled batches in one epoch, and validate the model at per epoch. Also, we pre-train the network on CityScapes [30] and finetune on KITTI [15], each for 200 epochs. Here we follow Eigen et al. [5]'s evaluation metrics for depth evaluation.

**Visual odometry prediction.**  For pose network, following Zhan et al. [17], we evaluate visual odometry results on KITTI odometry dataset [15], where sequence 00-08/09-10 are used for training/testing. We use the standard evaluation metrics by the dataset for trajectory evaluation rather than Zhou et al. [7]'s 5-frame pose evaluation, since they are more widely used and more meaningful.

### 4.2 Comparisons with the state-of-the-art

**Depth results on KITTI raw dataset.**  Tab. 1 shows the results on KITTI raw dataset [15], where our method achieves the state-of-the-art performance when compared with models trained on monocular video sequences. Note that recent work [9–11, 31] all jointly learn multiple tasks, while our approach does not. This effectively reduces the training and inference overhead. Moreover, our method competes quite favorably with other methods using stronger supervision signals such as calibrated stereo image pairs (*i.e.*, pose supervision) or even ground-truth depth annotation.

**Visual odometry results on KITTI odometry dataset.**  We compare with SfMLearner [7] and the methods trained with stereo videos [17]. We also report the results of ORB-SLAM [12] system (without loop closing) as a reference, though emphasize that this results in a comparison note between a simple frame-to-frame pose estimation framework with a Visual SLAM system, in which the latter has a strong back-end optimization system (*i.e.*, bundle adjustment [32]) for improving the performance. Here, we ignore the frames (First 9 and 30 respectively) from the sequences (09 and 10) for which ORB-SLAM [12] fails to output camera poses because of unsuccessful initialization.

Table 1: Single-view depth estimation results on test split of KITTI raw dataset [15]. The methods trained on KITTI raw dataset [15] are denoted by K. Models with pre-training on CityScapes [30] are denoted by CS+K. (D) denotes depth supervision, (B) denotes binocular/stereo input pairs, (M) denotes monocular video clips. (J) denotes joint learning of multiple tasks. The best performance in each block is highlighted as bold.

| Methods | Dataset | Error ↓ | | | | Accuracy ↑ | | |
|---|---|---|---|---|---|---|---|---|
| | | AbsRel | SqRel | RMS | RMSlog | $< 1.25$ | $< 1.25^2$ | $< 1.25^3$ |
| Eigen et al. [5] | K (D) | 0.203 | 1.548 | 6.307 | 0.282 | 0.702 | 0.890 | 0.958 |
| Liu et al. [6] | K (D) | 0.202 | 1.614 | 6.523 | 0.275 | 0.678 | 0.895 | 0.965 |
| Garg et al. [22] | K (B) | 0.152 | 1.226 | 5.849 | 0.246 | 0.784 | 0.921 | 0.967 |
| Kuznietsov et al. [18] | K (B+D) | **0.113** | **0.741** | **4.621** | **0.189** | **0.862** | **0.960** | **0.986** |
| Godard et al. [23] | K (B) | 0.148 | 1.344 | 5.927 | 0.247 | 0.803 | 0.922 | 0.964 |
| Godard et al. [23] | CS+K (B) | 0.124 | 1.076 | 5.311 | 0.219 | 0.847 | 0.942 | 0.973 |
| Zhan et al. [17] | K (B) | 0.144 | 1.391 | 5.869 | 0.241 | 0.803 | 0.928 | 0.969 |
| Zhou et al. [7] | K (M) | 0.208 | 1.768 | 6.856 | 0.283 | 0.678 | 0.885 | 0.957 |
| Yang et al. [31] (J) | K (M) | 0.182 | 1.481 | 6.501 | 0.267 | 0.725 | 0.906 | 0.963 |
| Mahjourian et al. [8] | K (M) | 0.163 | 1.240 | 6.220 | 0.250 | 0.762 | 0.916 | 0.968 |
| Wang et al. [16] | K (M) | 0.151 | 1.257 | 5.583 | 0.228 | 0.810 | 0.936 | 0.974 |
| Geonet-VGG [9] (J) | K (M) | 0.164 | 1.303 | 6.090 | 0.247 | 0.765 | 0.919 | 0.968 |
| Geonet-Resnet [9] (J) | K (M) | 0.155 | 1.296 | 5.857 | 0.233 | 0.793 | 0.931 | 0.973 |
| DF-Net [10] (J) | K (M) | 0.150 | 1.124 | 5.507 | 0.223 | 0.806 | 0.933 | 0.973 |
| CC [11] (J) | K (M) | 0.140 | **1.070** | **5.326** | **0.217** | 0.826 | 0.941 | **0.975** |
| Ours | K (M) | **0.137** | 1.089 | 5.439 | **0.217** | **0.830** | **0.942** | **0.975** |
| Zhou et al. [7] | CS+K (M) | 0.198 | 1.836 | 6.565 | 0.275 | 0.718 | 0.901 | 0.960 |
| Yang et al. [31] (J) | CS+K (M) | 0.165 | 1.360 | 6.641 | 0.248 | 0.750 | 0.914 | 0.969 |
| Mahjourian et al. [8] | CS+K (M) | 0.159 | 1.231 | 5.912 | 0.243 | 0.784 | 0.923 | 0.970 |
| Wang et al. [16] | CS+K (M) | 0.148 | 1.187 | 5.496 | 0.226 | 0.812 | 0.938 | 0.975 |
| Geonet-Resnet [9] (J) | CS+K (M) | 0.153 | 1.328 | 5.737 | 0.232 | 0.802 | 0.934 | 0.972 |
| DF-Net [10] (J) | CS+K (M) | 0.146 | 1.182 | 5.215 | 0.213 | 0.818 | 0.943 | **0.978** |
| CC [11] (J) | CS+K (M) | 0.139 | **1.032** | **5.199** | 0.213 | 0.827 | 0.943 | 0.977 |
| Ours | CS+K (M) | **0.128** | 1.047 | 5.234 | **0.208** | **0.846** | **0.947** | 0.976 |

Table 2: Visual odometry results on KITTI odometry dataset [15]. We report the performance of ORB-SLAM [12] as a reference and compare with recent deep methods. K denotes the model trained on KITTI, and CS+K denotes the model with pre-training on Cityscapes [30].

| Methods | Seq. 09 | | Seq. 10 | |
|---|---|---|---|---|
| | $t_{err}$ (%) | $r_{err}$ (°/100m) | $t_{err}$ (%) | $r_{err}$ (°/100m) |
| ORB-SLAM [12] | 15.30 | 0.26 | 3.68 | 0.48 |
| Zhou et al. [7] | 17.84 | 6.78 | 37.91 | 17.78 |
| Zhan et al. [17] | 11.93 | 3.91 | 12.45 | **3.46** |
| Ours (K) | 11.2 | 3.35 | **10.1** | 4.96 |
| Ours (CS+K) | **8.24** | **2.19** | 10.7 | 4.58 |

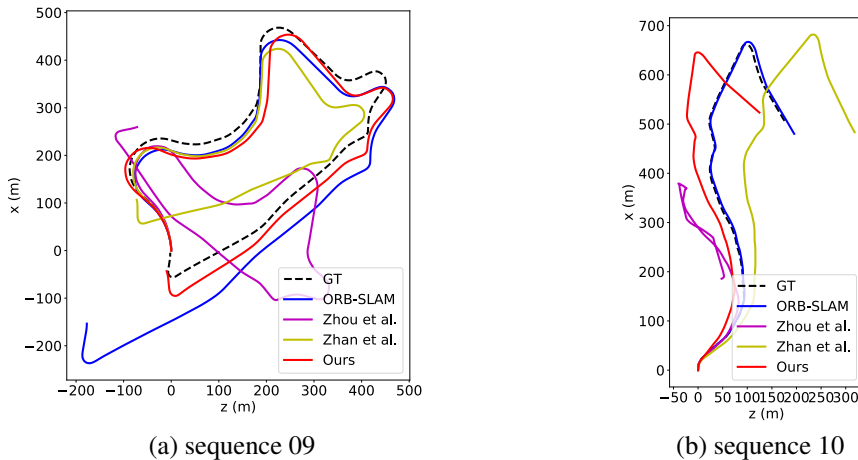

(a) sequence 09        (b) sequence 10

Figure 3: Qualitative results on the testing sequences of KITTI odometry dataset [15].

Tab. 2 shows the average translation and rotation errors for the testing sequence 09 and 10, and Fig. 3 shows qualitative results. Note that the comparison is highly disadvantageous to the proposed method: *i)* we align per-frame scale to the ground truth scale for [7] due to its scale-inconsistency, while we only align one global scale for our method; *ii)* [17] requires stereo videos for training, while we only use monocular videos. Although it is unfair to the proposed method, the results show that our method achieves competitive results with [17]. Even when compared with the ORB-SLAM [12] system, our method shows a lower translational error and a better visual result on sequence 09. This is a remarkable progress that deep models trained on unlabelled monocular videos can predict a globally scale-consistent visual odometry.

## 4.3  Ablation study

In this section, we first validate the efficacy of the proposed geometry-consistency loss $L_{GC}$ and the self-discovered weight mask $M$. Then we experiment with different scale numbers, network architectures, and image resolutions.

**Validating proposed $L_{GC}$ and $M$.**  We conduct ablation studies using DispNet [7] and images of $416 \times 128$ resolution. Tab. 3 shows the depth results for both single-scale and multi-scale supervisions. The results clearly demonstrate the contribution of our proposed terms to the overall performance. Besides, Fig. 4 shows the validation error during training, which indicates that the proposed $L_{GC}$ can effectively prevent the model from overfitting.

Table 3: Ablation studies on $L_{GC}$ and $M$. Brackets show results of multi-scale (4) supervisions.

| Methods | Error ↓ | | | | Accuracy ↑ | | |
| --- | --- | --- | --- | --- | --- | --- | --- |
| | AbsRel | SqRel | RMS | RMSlog | $< 1.25$ | $< 1.25^2$ | $< 1.25^3$ |
| Basic | 0.161 (0.185) | 1.225 | 5.765 | 0.237 | 0.780 | 0.927 | 0.972 |
| Basic+SSIM | 0.160 (0.163) | 1.230 | 5.950 | 0.243 | 0.775 | 0.923 | 0.969 |
| Basic+SSIM+GC | 0.158 (0.161) | 1.247 | 5.827 | 0.235 | 0.786 | 0.927 | 0.971 |
| Basic+SSIM+GC+M | **0.151** (**0.158**) | **1.154** | **5.716** | **0.232** | **0.798** | **0.930** | **0.972** |

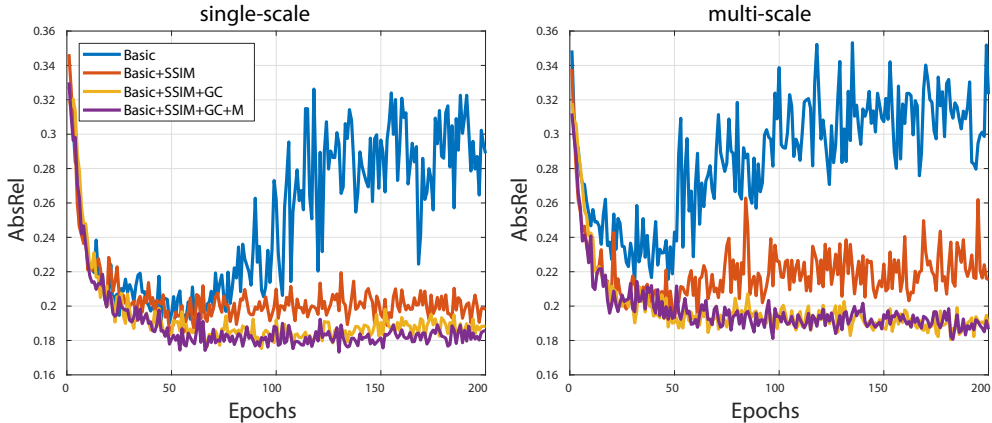

Figure 4: Validation error. Both *Basic* and *Basic+SSIM* overfit after about 50 epochs, while others do not due to proposed $L_{GC}$. Besides, models with the single-scale supervision in training outperforms those with multi-scale (4) supervisions.

**Proposed single-scale vs multi-scale supervisions.**  As mentioned in Sec. 4.1, we empirically find that using single-scale supervision leads to better performance than using the widely-used multi-scale solution. Tab. 4 shows the depth results. We hypothesis the reason is that the photometric loss is not accurate in low-resolution images, where the pixel color is over-smoothed. Besides, as the displacement between two consecutive views is small, the multi-scale solution is unnecessary.

**Network architectures and image resolutions.**  Tab. 5 shows the results of different network architectures on different resolution images, where DispNet and DispResNet are both borrowed from

Table 4: Ablation studies on scale numbers of supervision.

| | Error ↓ | | | | Accuracy ↑ | | |
|---|---|---|---|---|---|---|---|
| #Scales | AbsRel | SqRel | RMS | RMSlog | $< 1.25$ | $< 1.25^2$ | $< 1.25^3$ |
| 1 | **0.151** | **1.154** | **5.716** | **0.232** | **0.798** | **0.930** | **0.972** |
| 2 | 0.152 | 1.192 | 5.900 | 0.235 | 0.795 | 0.927 | 0.971 |
| 3 | 0.159 | 1.226 | 5.987 | 0.240 | 0.780 | 0.921 | 0.969 |
| 4 | 0.158 | 1.214 | 5.898 | 0.239 | 0.782 | 0.925 | 0.971 |

CC [11], and DispNet is also used in SfMLearner [7]. It shows that higher resolution images and deeper networks can results in better performance.

Table 5: Ablation studies on different network architectures and image resolutions.

| Methods | Resolutions | Error ↓ | | | | Accuracy ↑ | | |
|---|---|---|---|---|---|---|---|---|
| | | AbsRel | SqRel | RMS | RMSlog | $< 1.25$ | $< 1.25^2$ | $< 1.25^3$ |
| DispNet | $416 \times 128$ | 0.151 | 1.154 | 5.716 | 0.232 | 0.798 | 0.930 | 0.972 |
| DispResNet | | 0.149 | 1.137 | 5.771 | 0.230 | 0.799 | 0.932 | 0.973 |
| DispNet | $832 \times 256$ | 0.146 | 1.197 | 5.578 | 0.223 | 0.814 | 0.940 | **0.975** |
| DispResNet | | **0.137** | **1.089** | **5.439** | **0.217** | **0.830** | **0.942** | **0.975** |

## 4.4 Timing and memory analysis

**Training time and parameter numbers.** We compare with CC [11], and both methods are trained on a single 16GB Tesla V100 GPU. We measure the time taken for each training iteration consisting of forward and backward pass using a batch size of 4. The image resolution is $832 \times 256$. CC [11] needs train 3 parts, including (Depth, Pose), Flow, and Mask. In contrast our method only trains (Depth, Pose). In total, CC takes about *7 days* for training as reported by authors, while our method takes about *32 hours*. Tab. 6 shows the per-iteration time and model parameters of each network.

Table 6: Training time per iteration and model parameters for each network.

| | CC [11] | | | Ours |
|---|---|---|---|---|
| Network | (Depth, Pose) | Flow | Mask | (Depth, Pose) |
| Time | 0.96s | 1.32s | 0.48s | 0.55s |
| Parameter Numbers | (80.88M, 2.18M) | 39.28M | 5.22M | (80.88M, 1.59M) |

**Inference time.** We test models on a single RTX 2080 GPU. The batch size is 1, and the time is averaged over 100 iterations. Tab. 7 shows the results. The DispNet and DispResNet architectures are same with SfMLearner [7] and CC [11], respectively, so their speeds are theoretically same.

Table 7: Inference time on per image or image pair.

| | DispNet | DispResNet | PoseNet |
|---|---|---|---|
| $128 \times 416$ | 4.9 ms | 9.6 ms | 0.6 ms |
| $256 \times 832$ | 9.2 ms | 15.5 ms | 1.0 ms |

## 5 Conclusion

This paper presents an unsupervised learning framework for scale-consistent depth and ego-motion estimation. The core of the proposed approach is a geometry consistency loss for scale-consistency and a self-discovered mask for handling dynamic scenes. With the proposed learning framework, our depth model achieves the state-of-the-art performance on the KITTI [15] dataset, and our ego-motion network can show competitive visual odometry results with the model that is trained using stereo videos. To the best of our knowledge, this is the first work to show that deep models training on unlabelled monocular videos can predict a globally scale-consistent camera trajectory over a long sequence. In future work, we will focus on improving the visual odometry accuracy by incorporating drift correcting solutions into the current framework.

**Acknowledgments**

The work was supported by the Australian Centre for Robotic Vision, the Major Project for New Generation of AI (No. 2018AAA0100400), and NSFC (NO. 61922046). Jiawang would also like to thank TuSimple, where he started research in this field.

## Footnotes

[1]Monocular systems such as ORB-SLAM [12] suffer from the scale ambiguity issue, but their predictions are globally scale-consistent. However, recently learned models using monocular videos not only suffer from the scale ambiguity, but also predict scale-inconsistent results over different snippets.

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
