[Supplementary Material]

# SC-SfMLearner: Supplementary

**Jia-Wang Bian**[1,2], **Zhichao Li**[3], **Naiyan Wang**[3], **Huangying Zhan**[1,2]

**Chunhua Shen**[1,2], **Ming-Ming Cheng**[4], **Ian Reid**[1,2]

[1]University of Adelaide, Australia
[2]Australian Centre for Robotic Vision, Australia
[3]TuSimple, China
[4]Nankai University, China

## 1 Pose estimation results on 5-frame snippets

Although the visual odometry results shown in the main paper is more important, we also evaluate pose estimation results using Zhou et al. [1]'s evaluation metric on 5-frame snippets. Tab. 1 shows the results, where our method shows slightly lower performances with the state-of-the-art methods but the gap is small.

Table 1: Pose estimation results on KITTI odometry dataset.

|  | Seq. 09 | Seq. 10 |
|---|---|---|
| ORB-SLAM (full) | $0.014 \pm 0.008$ | $0.012 \pm 0.011$ |
| ORB-SLAM (short) | $0.064 \pm 0.141$ | $0.064 \pm 0.130$ |
| Mean Odometry | $0.032 \pm 0.026$ | $0.028 \pm 0.023$ |
| Zhou et al. [1] | $0.021 \pm 0.017$ | $0.020 \pm 0.015$ |
| Mahjourian et al. [2] | $0.013 \pm 0.010$ | $0.012 \pm 0.011$ |
| GeoNet [3] | $\mathbf{0.012 \pm 0.007}$ | $\mathbf{0.012 \pm 0.009}$ |
| DF-Net [4] | $0.017 \pm 0.007$ | $0.015 \pm 0.009$ |
| CC [5] | $\mathbf{0.012 \pm 0.007}$ | $\mathbf{0.012 \pm 0.009}$ |
| Ours | $0.016 \pm 0.007$ | $0.015 \pm 0.015$ |

## 2 Depth estimation results on Make3D dataset.

To verify the generalization ability of the trained model, we also test it on Make3D dataset [6]. Tab. 2 shows the relative depth error, where our model is trained on KITTI [7] without fine-tuning on Make3D [8]. The results demonstrate that our method performs slightly better than other state-of-the-art methods.

Table 2: Depth results (AbsRel) on Make3D [6] test set without finetuning.

| Methods | Zhou et al. [1] | Godard et al. [9] | DF-Net et al. [4] | CC [5] | Ours |
|---|---|---|---|---|---|
| AbsRel | 0.383 | 0.544 | 0.331 | 0.320 | **0.312** |

## 3 More qualitative results

Fig. 1 illustrates visual results of depth estimation and occlusion detection by the proposed approach. It demonstrates the efficacy of proposed mask in terms of detecting moving objects and occlusions.

Figure 1: Visual results. Top to bottom: sample image, estimated depth, self-discovered mask. The proposed mask can effectively identify inconsistent pixels caused by moving objects and occlusions.