[Reviews · NeurIPS 2019]

Reviewer 1



Overview: The paper proposes a geometric constraint that allows to train ego-motion and depth networks on monocular videos. While the resulting approach is relatively simple, it is very efficient and achieves state of the art or near state of the art results. The authors provide extensive ablation studies and simulations to complement their work. Quality: The quality of this contribution is generally high. Most notably, authors provide extensive ablation studies and use simulations to elucidate the properties of the proposed method. Clarity: The paper is clearly written and is easy to read. Originality: The proposed method is certainly novel and the contribution is original enough. Impact: While other alternative exist that may allow to achieve similar performance, the proposed method is simple, efficient, and does not require expensive-to-obtain datasets. As such, it may prove very useful in practical applications. Conclusion: This paper tackles an important problem and proposes an original solution that may be useful in practical settings. The quality of the experiments is high. Overall, this paper is above the acceptance threshold. # In their response, the authors provided additional details and clarified what they meant by efficiency (mainly training time efficiency). In the discussion, R3 mentioned a particular concern that while efficiency is very important for this contribution, it receives relatively little attention in the article. While it is partially addressed in the rebuttal, I believe that it would be beneficial to further elaborate on how much training time different systems take in previous methods, as it is not entirely clear at the moment and requires re-reading specific aspects of previous SOTA works. In particular, it is important to clarify the phrase "iterative training" the authors used in the response. I.e. it would help to delineate what does every iteration consist of, how many iterations are usually required, how much time each iteration takes, etc. Overall, I agree that these concerns are important, but I believe they could be significantly amended without additional experimentation, by slightly rewriting/expanding relevant parts of the paper. I believe that the authors will be able to do so before the final submission. Moreover, while important, I don't think that those concerns are critical, as they are mostly about how results are presented, not about the results themselves. Therefore, I keep my score unchanged.

Reviewer 2



The originality of the method stems from the combination of a number of known reasonable techniques. Related work is cited well and is clearly differentiated from the proposed method. The visual odometry results seem highly encouraging, both qualitatively and quantitively, but I am not very familiar enough with the related work to be certain. It's not clear how good the KITTI depth results are. For the non-pretrained setting, the proposed method is only marginally better than the cited CC method [10], which as a joint learning method incorporates weaker priors compared to the strong priors required for the method proposed by the authors. The paper is written very clearly and because the proposed method combines rather simple ingredients it is easy to follow along. Overall I think this is a fair submission, it's good to have empirical data that the proposed methods (in particular the geometry consistency) do in fact help. The authors did not significantly exceed state of the art results for depth estimation except for special cases. However for future research, the proposed method can plausibly be combined with others methods to move the state of the art forward, and quantitatively they already provide an improvement for visual odometry tasks. UPDATE: After reading the authors' rebuttal, I have chosen to maintain my score, for to the following reasons: (1) the authors address the concern about the strength of the priors and this is a helpful component of the rebuttal, (2) the authors helpfully explain the training time improvements, as other reviewers have noted, this should be made clearer in the paper and discuss both training and test time differences between the proposed method and prior work. However, (3) the remaining main flaw of the paper is that the contributions are of theoretical and efficiency nature, it would be more convincing to see the impact of such contributions as an increase in final accuracy. For example, this could be shown with a comparison of accuracy between the proposed and other methods after similar amounts of training time/compute.

Reviewer 3



The paper describes an elegant methodology to improves the performance of the very important and difficult task of depth and ego-motion estimation from monocular video. Experimental results validate the efficacy of the proposed methods. The paper is well organized and well written. The paper would benefit from a more thoroughly study of the proposed methods: A mask for the photometric loss is proposed as a more efficient a simpler alternative to estimating optical flow. A natural question is how many milliseconds of GPU time does this save which isn't address.

[Author Response · NeurIPS 2019]

We thank all reviewers for their efforts and positive comments.

To Review #1: Thank you for the comments and we will follow the suggestions to do more experiments and analyses.

To Review #2: Thank you very much for confirming our contributions.

Q1) [Strong priors]: "geometry-consistency" is not a manually designed prior for specific scenes. It is indeed an
underlying constraint of Structure-from-Motion or Multi-View Geometry [3]. A well-known method that enforces
geometry-consistency is Bundle Adjustment. BA is the core method of Visual SLAM and SfM, and it is widely used in
general scenes. Here, unsupervised depth learning relies on the theory of Structure-from-Motion, as discussed in [21],
so the baseline research on this problem is called SfMLearner [6]. CC [10] and all other related works share the same
theory for unsupervised depth learning. Therefore, compared with them, our method does not require additional strong
priors. Our contribution is more deeply leveraging SfM theory in unsupervised depth learning.

Q2) [Improvement over SOTA]: Yes, although the performance is just slightly better than CC [10], our contribution is
solving scale-inconsistency issue and a more efficient framework. More advantages like efficiency would be added.

To Review #3: Thank you for your comments.

Q3) [How much cost is saving]: We measure the time taken for each iteration consisting of forward and backward pass
using a batch size of 4. Below shows the results on $832 \times 256$ images. CC [10] needs train 3 parts iteratively, including
(DepthNet, PoseNet), FlowNet, and MaskNet. Our method only trains (DepthNet, PoseNet) for 200K iterations. In
total, CC takes about **7 days** for training, while our method takes **32h27m10s**. The results of CC [10] are reported by
authors. Both methods are tested on a single 16GB Tesla V100 GPU.

Table 1: Training time and the number of model parameters for each network

|  | CC [10] | | | Ours |
| --- | --- | --- | --- | --- |
| Network | (DepthNet, PoseNet) | FlowNet | MaskNet | (DepthNet, PoseNet) |
| Time | 0.96s | 1.32s | 0.48s | 0.55s |
| Parameter Numbers | (80.88M, 2.18M) | 39.28M | 5.22M | (80.88M, 1.59M) |

Moreover, we show the time consumption for obtaining mask. CC involves FlowNet and MaskNet, and we use GC+M.
Time for one iteration is shown below. FlowNet is slow due to correlation calculation, which is time-consuming.
Ours (GC+M) is fast because only several basic operations such as subtraction and division are involved for mask
computation, as indicated in Eqn 5-8. Note that the most time-consuming part (reprojection) in our method has been
done when computing photometric loss, which can be reused in computing mask.

Table 2: Time to obtaining mask on a single 8GB RTX 2080.

|  | CC [10] | | Ours |
| --- | --- | --- | --- |
| Method | FlowNet (Forward) | MaskNet (Forward) | GC+M |
| Time | 42.73ms | 16.94ms | 0.0002ms |

Q4) [Moving objects, Static scene assumption]: Static assumption (or called rigid assumption) is the prerequisite of
perspective projection [3], i.e., depth and pose based projection. The perspective projection allows for photometric loss
in unsupervised depth learning. Therefore, photometric loss on dynamic scenes (moving objects) are meaningless and
cause false supervisions. The rigid assumption and perspective projection were discussed in Mult-View Geometry [3].
Besides, GeoNet[8], DF-Net[9], and CC[10] introduced the adverse effect of moving objects for unsupervised depth
learning and used optical flow networks to localize dynamic regions.

Q5) [Scale ambiguity, Inconsistency issue]: Scale ambiguity is a well-recognized issue in pure monocular systems. It
refers to the issue that the scale that aligns the estimated scene to the real-world scene is unknown. It can be resolved by
introducing another devices (stereo camera or range sensor) or relying on specific strong priors. Indeed, all related
works (from SfMLearner to CC, and ORB-SLAM) estimate relative depths instead of real depths due to scale ambiguity.
Inconsistency issue arises due to the fact that SfMLearner and related methods take one snippet (3 or 5 frames) as input
during tracking and do not consider the consistency between different snippets. As a result, the scaling factors that align
the estimated camera poses (translation) to real-world varies for each snippet, i.e. inconsistent scales. Therefore, related
works evaluate pose estimation results using 5-frame pose metrics [6] instead of using more general visual odometry
evaluation metrics. In our method, we enforce consistent scales by enforcing depth consistency across snippets, which
further allows pose consistency across snippets.

Q6) [Mask mitigates the issue of moving objects]: Fig 2 demonstrates that our mask can detect moving objects and
occlusions. More visual results are in supplementary. Tab 1 demonstrates that using mask can boost performance.

Q7) [Compare with [7]]: Have compared in Tab 3. Regarding AbsRel, it is 0.137 vs 0.163 in (K) and 0.128 vs 0.159 in
(CS+K).

[Meta-Review · NeurIPS 2019]

This paper generated a large amount of discussion. On the positive side, the reviewers noted that the paper shows large benefits compared to baseline approaches in terms of producing scale-consistent outputs and significantly reduced training time. The reviewers appreciated the clarification on the phrase "efficiency" as explained in the rebuttal, as well as other clarifications about priors. For the final revision, we strongly recommend that the authors: - Describe in more detail why their method leads to faster training times, compared to the previous methods; this was a very confusing point to some of the reviewers - Explain the positive benefits of reduced training time, e.g. lowering the computational resources needed for training and thereby facilitating reuse of the method - Compute inference efficiency, i.e. runtime of the trained model at test time, compared to baseline approaches